# Fructosyl Amino Oxidase as a Therapeutic Enzyme in Age-Related Macular Degeneration

**DOI:** 10.3390/ijms25094779

**Published:** 2024-04-27

**Authors:** Joris R. Delanghe, Jose Diana Di Mavungu, Koen Beerens, Jonas Himpe, Nezahat Bostan, Marijn M. Speeckaert, Henk Vrielinck, Anne Vral, Caroline Van Den Broeke, Manon Huizing, Elisabeth Van Aken

**Affiliations:** 1Department of Diagnostic Sciences, Ghent University, 9000 Ghent, Belgium; jonas.himpe@ugent.be; 2Department of Green Chemistry and Technology, MSsmall Expertise Centre, Mass Spectrometry Analysis of Small Organic Molecules, Ghent University, 9000 Ghent, Belgium; jose.dianadimavungu@ugent.be; 3Department of Biotechnology, Faculty of Bioscience Engineering, Ghent University, 9000 Ghent, Belgium; koen.beerens@ugent.be; 4Antwerp Biobank, Antwerp University Hospital, 2650 Antwerp, Belgium; nezahat.bostan@uza.be (N.B.); manon.huizing@uza.be (M.H.); 5Department of Internal Medicine and Pediatrics, Ghent University, 9000 Ghent, Belgium; marijn.speeckaert@ugent.be; 6Department of Solid State Sciences, Ghent University, 9000 Ghent, Belgium; henk.vrielinck@ugent.be; 7Department of Human Structure and Repair, Ghent University, 9000 Ghent, Belgium; anne.vral@ugent.be; 8Department of Pathology, Ghent University Hospital, 9000 Ghent, Belgium; caroline.vandenbroecke@azstlucas.be; 9Department of Head and Skin, Ghent University, 9000 Ghent, Belgium; elisabeth.vanaken@ugent.be

**Keywords:** ageing, age-related macular degeneration, fructosyl amino oxidase, glycation

## Abstract

Age-related macular degeneration (AMD) is an age-related disorder that is a global public health problem. The non-enzymatic Maillard reaction results in the formation of advanced glycation end products (AGEs). Accumulation of AGEs in drusen plays a key role in AMD. AGE-reducing drugs may contribute to the prevention and treatment of AGE-related disease. Fructosamine oxidase (FAOD) acts on fructosyl lysine and fructosyl valine. Based upon the published results of fructosamine 3-kinase (FN3K) and FAOD obtained in cataract and presbyopia, we studied ex vivo FAOD treatment as a non-invasive AMD therapy. On glycolaldehyde-treated porcine retinas, FAOD significantly reduced AGE autofluorescence (*p* = 0.001). FAOD treatment results in a breakdown of AGEs, as evidenced using UV fluorescence, near-infrared microspectroscopy on stained tissue sections of human retina, and gel permeation chromatography. Drusen are accumulations of AGEs that build up between Bruch’s membrane and the retinal pigment epithelium. On microscopy slides of human retina affected by AMD, a significant reduction in drusen surface to 45 ± 21% was observed following FAOD treatment. Enzymatic digestion followed by mass spectrometry of fructose- and glucose-based AGEs (produced in vitro) revealed a broader spectrum of substrates for FAOD, as compared to FN3K, including the following: fructosyllysine, carboxymethyllysine, carboxyethyllysine, and imidazolone. In contrast to FN3K digestion, agmatine (4-aminobutyl-guanidine) was formed following FAOD treatment in vitro. The present study highlights the therapeutic potential of FAOD in AMD by repairing glycation-induced damage.

## 1. Introduction

In the elderly, age-related macular degeneration (AMD) is a common vision disorder. In industrialized countries, AMD is the most common cause of severe loss of eyesight among people aged 50 and older and is considered globally as a growing public health problem [1]. At present, only the wet form of AMD (representing only about 10% of the AMD patients) can be treated [2].

Aging is considered as the most important risk factor, with related damage due to free radicals being a major pathogenic component [3]. In the slow process of protein glycation, reducing sugars and carbonyls react with free amino groups, forming adducts that subsequently rearrange and react further, eventually leading to the formation of protein cross-links. This very complex set of non-enzymatic reactions is known as the Maillard reaction [3,4]. A Schiff base is formed in the first steps of the Maillard reaction, whereby a reducing sugar (e.g., glucose) reacts with a protein-bound amino group. Labile Schiff bases may lead to the formation of reactive dicarbonyls (e.g., methylglyoxal) or may lead to the formation of stable Amadori compounds such as carboxymethyllysine (CML) and carboxyethyllysine (CEL). The advanced stages of the Maillard reaction eventually lead to the in vivo formation of stable protein-cross-links and adducts, or advanced glycation end products (AGEs) [5,6,7,8].

The role of glycated proteins in the ageing retina and AMD is well known [9]. Endogenous and exogenous advanced glycation end products (AGEs) slowly accumulate with age in the outer retina and thickened Bruch’s membrane [9,10,11]. Proteome analysis has demonstrated an abundance of cross-linked proteins in the drusen of AMD patients, compared to in a typical Bruch’s membrane [3,12]. Retinal pigment epithelium (RPE) lipofuscin consists of a complex mixture of bisretinoid fluorophores that have been identified using chromatography and mass spectrometry and have been structurally characterized as AGEs. These fluorescent AGEs amass in healthy RPE cells, but more so in early and advanced dry AMD compared to age-matched controls [13]. The spectral characteristics of fundus autofluorescence are consistent with those of RPE lipofuscin, predominantly with an origin from the fluorescent AGEs [13]. Ex vivo, RPE lipofuscin exhibits an excitation spectrum that peaks between 450 and 490 nm [14].

The progressive accumulation of AGEs caused by the Maillard reaction may contribute to general aging [5,6,7,8]. The non-enzymatic glycation of long-lived proteins is responsible for altering protein structure and stability and for inducing the covalent cross-linking, aggregation, and insolubilization of proteins, which results in the thickening of the Bruch’s membrane [12]. In aging, increased concentrations of dicarbonyl compounds (e.g., methylglyoxal and glyoxal) also result in AGE cross-links [4]. AGE-inhibiting or -disrupting compounds may have efficacy in the prevention and treatment of AGE-related processes. Recently, the related human deglycating enzyme fructosamine 3-kinase (FN3K) has been shown to reverse glycation in cataract and AMD [15].

The bacterial enzyme fructosamine oxidase (EC Fructosyl-amino acid oxidase (FAOD); fructosyl-α-l-amino acid: oxygen oxidoreductase (defructosylating)), is an enzyme present in yeast and many bacteria [16,17,18]. FAOD catalyzes the oxidation of the C-N bond linking the C1 of the fructosyl moiety and the nitrogen of the amino group of fructosyl amino acids. Flavin adenine dinucleotide (FAD) acts as its cofactor. FAOD-based methods for assaying glycated proteins have been used for in vitro diagnostics applications. However, intact hemoglobin A1c is unable to react with FAOD and, thus, specimens require a preparative proteolytic digestion step to liberate glycated amino acids or glycated dipeptides [19]. As a result, the therapeutic use of FAOD in AGEs-induced conditions in animals or humans has only recently been considered in the treatment of glycation-associated disorders such as presbyopia and hypertrophic scars [20,21].

As the crosslinks encountered are mainly non-disulfide bridges, and based upon the earlier positive results of deglycating enzymes in reversing presbyopia and cataract (other eye diseases caused by advanced protein glycation of the lens), we explored, ex vivo, the effects of topical deglycating enzyme (FAOD) treatment on the retina as a new potential non-invasive treatment for AMD. In order to achieve this goal, a combination of analytical techniques (light microscopy, near infrared microscopy, ultraviolet fluorescence, gel permeation chromatography, and mass spectrometry) will be used. Furthermore, we compared the catalytic action of FAOD (substrate specificity and molecular mass of formed AGEs) with the earlier studied deglycating enzyme fructosamine 3 kinase (FN3K) [15]. As the availability of human retina tissue affected by AMD is limited (cadaver eyes), we have carried out additional experiments on porcine retinas to expand the amount of experiments and data.

## 2. Results

### 2.1. Autofluorescence Kinetics of FAOD Treatment on Retina Suspensions

Figure 1 shows the mean change of AF values of glycolaldehyde-treated and post-glycation FAOD-treated retinas compared to baseline levels for human retina suspensions.

Glycolaldehyde-treated porcine retinas showed a significant ultraviolet autofluorescence (440 nm) reduction following FAOD treatment; on average, a 43% ± 4% decrease in autofluorescence was observed (*p* = 0.001). Treatment of the retinas with the non-active FAOD mutant enzyme did not result in a significant effect. (not shown).

### 2.2. Human Retina

Near-infrared microspectroscopy on stained tissue sections of human retina treated with FN3K and FAOD were compared. Spectral changes were observed (Figure 2). The obtained Hotelling plot shows a clear distinction between FN3K-treated drusen and FAOD-treated drusen (Figure 3). Following FAOD treatment, mean surface of drusen reduced to 45 ± 21% of the initial surface (*p* < 0.005) (Appendix A).

### 2.3. Comparison between FAOD and F3K

Following incubation of arginine or lysine with glucose, a broad variety of lysine (Table 1, Figure 4) and arginine (Table 2, Figure 5) compounds are spontaneously formed. Following the treatment of AGEs with either FAOD or F3K, enzymatic digests of AGEs (obtained via the co-incubation of sugars and amino acids) were subjected to mass spectrometric analysis. The resulting spectra were compared. The results are reported in Table 3. Following FAOD treatment, the arginine-derived compounds ornithine, fructosylarginine/glucosylarginine, and imidazolone A showed a strong disappearance, whereas F3K showed only a weak result (arginine) or proved to be non-effective. Furthermore, the lysine-based AGEs fructosyllysine, carboxyethyl (CEL), and carboxymethyllysine (CML) disappeared from the AGE mixture following FAOD treatment, as compared to the parallel digestion by F3K. As a result of FAOD activity, agmatine (4-aminobutyl-guanidine) was formed. In the case of FN3K, the amount of formed agmatine was lower.

The gel filtration patterns (Sephadex G-25^®^ gel, Cytiva, Uppsala, Sweden) of AGEs revealed the presence of high molecular mass compounds, which disintegrated after FAOD treatment. In parallel, AF was reduced following treatment with the FAOD enzyme. As compared to the fragments obtained following FN3K digestion, fragments obtained following FAOD digestion are similar in size (on average, 3.5 to 4 kDa) (Figure 6).

## 3. Discussion

In vitro FAOD treatment of porcine AGE-modified retina showed a significant reduction in AGEs. Similarly, AGEs present in human drusen were significantly reduced following FAOD treatment, as evidenced using both ultraviolet autofluorescence (440 nm) reduction and spectral changes on infrared microscopy.

In agreement with earlier observations on FN3K, our observations strengthen our hypothesis that the process of non-enzymatic glycation of long-lived proteins can be reversed [15,20,21,22]. The absence of any chemical effect following administration of the inactive mutant FAOD enzyme proves that the observed changes following FAOD treatment are due to the catalytic activity of the FAOD, per se.

The physicochemical behavior of deglycating enzyme molecules (like FAOD (49 kDa) [17] and fructosamine 3 kinase (37 kDa)) [23] allows swift diffusion of these enzymes in the anterior and posterior eye chamber, so that intraocular targets (e.g., the human retina) are within reach, when considering topical treatment (eye drops) [23].

In contrast to FN3K, no cofactors like magnesium or ATP are required in large concentrations, to maintain the FAOD enzyme activity. Flavin adenine dinucleotide (FAD, the cofactor of FAOD) is abundant in human biological fluids, so that the addition of FAD is not required for therapeutic purposes. The reported concentrations of FAD are about 0.34–0.63 μg mL^−1^ in cells [24] and 0.04–0.06 μg mL^−1^ in human plasma [24], respectively. Furthermore, FAOD contains less vulnerable sulfhydryl groups and is, therefore, less prone to oxidation, as compared to FN3K. Gel filtration patterns of AGEs revealed the presence of high molecular mass compounds, which disintegrated after FAOD treatment. These experiments show that FAOD treatment results in the destruction of cross-links and a deglycation of macro-molecular AGEs. As compared to the fragments obtained following FN3K digestion, fragments obtained following FAOD digestion are similar in size (on average, 3.5 to 4 kDa).

Our preliminary ex vivo data might be promising for the pharmaceutical treatment of AMD in a cost-effective way. The effects of FAOD on drusen are expected to be long-lasting, since the rate of (re)glycation is slow [23,25,25]. Since enzymes are characterized by a high turnover rate (the maximal number of substrate molecules converted to product per active site per unit time), the required therapeutic FAOD activity is estimated to be extremely low. The low FAOD concentration in the eye drops (~2 µmol/L) yield intraocular FAOD concentrations in the nmol/L range, which minimizes the chances for adverse effects. Also, intravitreal FAOD injections could be considered for AMD treatment. Although, in contrast to FN3K (a recombinant human enzyme), FAOD is a microbial enzyme, it should be taken into account that the human eye is generally considered to be an immune-privileged organ, in which the expected immunological side effects of exposure to small amounts of non-human proteins are small [26].

A mass spectrometry-based comparison of F3K and FAOD enzymatic digests of common AGEs demonstrated that FAOD is able to destroy a broader spectrum of AGEs, as compared to FN3K. Both CML and CEL are catabolized by FAOD action. The latter two most abundant AGEs have been identified as AGES in the Bruch’s membrane of ageing retinas [27]. Also, imidazolone (another FAOD substrate) has been identified as an AGE, occurring in ageing retinas [28]. Oxidative protein modifications like CML are elevated in AMD Bruch’s membranes and stimulate neovascularization in vivo (advanced wet AMD) [29], suggesting possible roles in choroidal neovascularization.

Our study is hampered by a number of limitations. First of all, experiments in this study have only been performed on in vitro or ex vivo material. Human clinical trials are indispensable for assessing the clinical validity of our findings. It can be anticipated that for the in vivo situation, in order to obtain optimal results, multiple treatment rounds will be needed. Besides, it might as well be that, after a certain amount of time in humans, protein crosslinking recurs and treatment should be repeated. Finally, the power of our study is hampered by a low number of human eyes.

Overall, it can be concluded that the FAOD enzyme treatment represents a potential treatment option for AMD. While our preliminary data are in need of further validation on larger sample sizes and need to be confirmed by human clinical trials, this study paves the way for future research on therapeutic deglycating enzymes.

## 4. Materials and Methods

### 4.1. FAOD and FAOD Mutant

Recombinant FAOD from Cryptococcus neoformans (0.45 U/mg protein) was purchased from Creative enzymes (Shirley, NY, USA). The enzyme was aliquoted, snap-frozen in liquid nitrogen, and stored at −80 °C. In parallel, an E280L mutant of the *Aspergillus fumigatus* FAOX-II (PDB code 3DJE = UniProt ID: P78573) was produced [17], which was demonstrated to be enzymatically non-active.

#### 4.1.1. Retina Material

Human retinas were prepared from cadaver eyes (n = 2) that had been rejected for corneal transplantation (Biobank Antwerpen, Antwerp, Belgium, ID71030031000). Retinas were isolated through dissection by a trained ophthalmologist within 12 h post mortem and were immediately transferred to a sterile 6-well plate and stored at 4 °C in RPMI-1640 medium (Sigma-Aldrich, St. Louis, MO, USA). The experiment was initiated within 48 h, by removing the RPMI medium and carefully washing the retinas with PBS. Subsequently, fluorescence readings were performed at baseline at 30 different retinal locations on each retina, with a fixed distance and 90° angle.

Porcine eyes (*n* = 20) were obtained from a local slaughter house and stored at 4 °C until processing. Neural retinas were prepared by a trained ophthalmologist through dissection within 12 h post mortem, transferred to a sterile 6-well plate (Thermo scientific, Roskilde, Denmark), and frozen (−20 °C). Subsequently, retina fragments were cut and added to a well of a black 96-well plate for recording the fluorescence (FluoroNunc PolySorp, Thermo Fisher Scientific, Waltham, MA, USA). Finally, baseline fluorescence was recorded for each retinal fragment at a fixed distance and 90° angle.

Since glycolaldehyde is a component to modify proteins via AGE formation and has a proven role in AMD pathogenesis [20], AGE modification was performed via the incubation of retinal fragments with 200 µL of 25 mm glycolaldehyde dimer (crystalline form, Sigma-Aldrich) in phosphate-buffered saline (PBS) at 37 °C for 3 h. After incubation, the active agents were carefully washed out. Retina fragments were subsequently stored overnight at 4 °C until termination of the chemical reaction.

Afterwards, in vitro deglycation was initiated using a solution containing 1.6 µg/mL FN3K in PBS. Similarly, incubation for 20 h at 37 °C in 2 mL of a solution containing 3.84 U/mL FAOD in 0.1 mL PBS was carried out. After the procedure, all wells were washed with PBS and fluorescence was re-measured. After incubation, the retina was washed five times with PBS and fluorescence measurements were repeated. Retinas were incubated with FAOD, mutant enzyme, or PBS.

#### 4.1.2. Infrared Spectroscopy of Human Retina

Near-infrared microspectroscopy on stained tissue sections of human retina with AGEs demonstrates different biochemical changes after FN3K treatment, compared to FAOD treatment. Donor eyes were obtained from two patients with stage 3 AMD (age > 70 years). After tissue sectioning, samples were deparaffinized prior to treatment through consecutive submerging in xylene (3 × 1.5 min), alcohol (90% 2 × 1 min; 75% 1 × 1 min), and rinsing in water. Slides were dried at 60 °C for 10 min. For the control treatment, one section was covered with 1 mL of ATP/MgCl2 solution. For FN3K treatment, an adjacent section was treated using 1 mL of FN3K solution (FN3K 250 µg/mL + ATP 5 mmol/L + MgCl_2_ 2 mmol/L. For FAOD treatment, an adjacent section was treated using 1 mL of FAOD solution 3.83 U/mL. The sections were incubated for 24 h at 37 °C. After incubation, tissue sections were carefully washed with distilled water and dried overnight at 37 °C. Sections were then stained and cover-slipped. Infrared (IR) microspectroscopy combines light microscopy with IR spectroscopy and is a powerful analytical method to obtain biochemically selective visualizations of tissue sections [30,31]. IR spectroscopy uses the principle that different regions of IR light are absorbed by various molecules within tissues (e.g., proteins, carbohydrates, and lipids) [31,32,33]. In a typical IR microspectroscopy system, visible light is employed to visualize and target the zones of interest on tissue sections. Once that particular area is found (e.g., a specific drusen), the system switches to the IR configuration and IR light is focused onto the predefined target [32]. To obtain a chemical fingerprint of drusen lesions on the identical tissue sections used for light microscopic examination, Fourier transform near-infrared (FT-NIR) transmission microspectra were studied with a Bruker Hyperion 2000 microscope coupled to a Bruker Vertex 80v FTIR spectrometer (Bruker, Billerica, MA, USA) that was equipped with a halogen light source, a CaF2 beam splitter, and an InGaAs detector. The aperture of the microscope was set at 50 µm × 50 µm and the objective magnification of the microscope at 15×. The background was collected with 800 co-adds. Spectra were recorded at a resolution of 16 cm^−1^ in the range from 12,000 to 4000 cm^−1^ (800 scans). Spectral data analysis was carried out using SIMCA software version 15.0 (MKS Data Analytics Solutions, Malmö, Sweden). Various preprocessing steps were performed to standardize the spectroscopic signals and to minimize irrelevant light scatter. Differentiation was carried out to reduce baseline effects and to accentuate small structural differences [32]. Standard normal variate normalization (SNV) was used to eliminate additive baseline offset variations and multiplicative scaling effects. Following preprocessing, spectral data were further analyzed using unsupervised pattern recognition methods, such as principal component analysis (PCA), and supervised pattern recognition methods, such as partial least squares-discriminant analysis (PLS-DA).

#### 4.1.3. Autofluorescence Measurement of AGEs

AGEs were assayed based on Maillard-type autofluorescence (AF) measurements (excitation 365 nm, emission 390–700 nm) using a Flame miniature spectrometer (FLAME-S-VIS-NIR-ES, 350–1000 nm, Ocean Optics, Dunedin, FL, USA) in combination with a high-power LED light source (365 nm, Ocean Optics) and a reflection probe (QR400-7-VIS-BX, Ocean Optics). The readings were averaged over 128 scans. AF values were calculated by dividing the average light intensity emitted per nm in the wavelength range 407–677 nm by the average light intensity per nm in the 342–407 nm range.

#### 4.1.4. Gel Permeation Chromatography

The AGE mixture was incubated (3 h, 37 °C) in 200 µL of an FAOD-containing solution (3.83 U/mL). Gel permeation chromatography of FN3K- and FAOD-treated AGEs was carried out on a column (length: 60 cm, inner diameter: 15 mm) packed with Sephadex G-25^®^ Fine resin (Sigma-Aldrich), in order to assess the apparent molecular mass of fructose-containing compounds (e.g., AGEs). Following fractionation, the presence of AGEs was first demonstrated based on Maillard-type autofluorescence (AF) measurements (excitation 365 nm, emission 390–700 nm) using a Flame miniature spectrometer (FLAME-S-VIS-NIR-ES, 350–1000 nm, Ocean Optics, Dunedin, FL, USA) equipped with a high-power LED light source (365 nm, Ocean Optics) and a reflection probe (QR400-7-VIS-BX, Ocean Optics). Autofluorescence peaks for FAOD-treated AGEs were detected at a wavelength of 520 nm. All obtained fractions were photometrically measured using the resorcinol–HCl (Seliwanoff) reaction, a classical color reaction for the presence of ketoses [34]. A 50 µL sample was added to 100 µL resorcinol (9 mM, Sigma-Aldrich) and 1 mL hydrochloric acid (9 mol/L, Sigma-Aldrich). Following a 5 min incubation in a boiling water bath, the color development was read photometrically at 488 nm in a 10 mm cuvette.

#### 4.1.5. UHPLC-HRMS Analysis of AGEs

The aim of this study was to obtain an initial insight into the nature of the compounds that are formed upon incubation of mixtures of amino acids (lysine or arginine) and sugars (glucose or fructose), followed by deglycation enzyme treatment (FAOD or FN3K). An untargeted metabolite profiling approach based on ultra-high performance liquid chromatography (UHPLC) coupled with high-resolution mass spectrometry (HRMS) was applied.

Solutions of glucose, fructose, arginine, and lysine were provided for the purpose of optimization and the study of the MS fragmentation pattern of these compounds. Amino acid–sugar mixtures, prior to enzyme treatment (group 1), were compared to various amino acid–sugar mixtures, such as the following: mixture of glucose (100 mg/mL) and arginine (100 mg/mL) in water, incubated at 37 °C for one week; mixture of fructose (100 mg/mL) and arginine (100 mg/mL) in water, incubated at 37 °C for one week; mixture of glucose (100 mg/mL) and lysine (100 mg/mL) in water, incubated at 37 °C for one week; and a mixture of fructose (100 mg/mL) and lysine (100 mg/mL) in water, incubated at 37 °C for one week.

After one week, the mixtures containing AGES were digested with either FAOD (3.83 U/L) or FN3K (FN3K (WO2019149648) 250 µg/mL + ATP 5 mmol/L + MgCl_2_ 2 mmol/L.

Mixtures of group 1 were diluted 200-fold, prior to UHPLC-HRMS analysis, while mixtures of group 1 were used as they were provided. All samples were filtered through centrifugation at 6000 rpm for 10 min, using an Ultrafree-MC centrifugal device (Millipore, Burlington, MA, USA).

Chromatographic separation was achieved on an Accela 1250 pump (Thermo Fisher Scientific), using a Zorbax RRHD Eclipse Plus reverse-phase C18 column (100A, 1.8 μm, 100 mm × 2.1 mm). The mobile phase consisted of 0.1% (*v*/*v*) formic acid in water (eluent A) and 0.1% (*v*/*v*) formic acid in methanol (eluent B). A gradient elution program was applied as follows: 0–0.5 min: 5% B, 0.5–20.0 min: 5–99% B, 20.0–21.0 min: 99% B, 21.0–24.0 min: 99–5% B, 24.0–28.0 min: 5% B. The mobile phase flow rate was 0.3 mL/min. The column temperature was set at 40 °C and the temperature of the autosampler was 10 °C. The injection volume was 5 µL.

High-resolution accurate mass and tandem mass spectrometry (MS/MS) fragmentation data were obtained using a Q-Exactive hybrid quadrupole-Orbitrap mass spectrometer (Thermo Fisher Scientific) equipped with a with heated-electrospray ionization (HESI-II) interface. The instrument was operated in the positive ionization mode. Data acquisition included full MS and data-dependent MS/MS scans. The ionization source parameters were as follows: a spray voltage of 3.0 kV, a capillary temperature of 350 °C, a heater temperature of 375 °C, a sheath gas flow rate of 45 arbitrary units (a.u.), and an auxiliary gas flow rate of 10 a.u. Daily external calibration of the HRMS was performed using the Calmix solution from Thermo Scientific, over a mass range of 138–1721 Da. Online mass calibration using diisooctyl phthalate (C_24_H_38_O_4_) as a lock mass was enabled.

Instrument control was carried out using Xcalibur 4.2 software (Thermo Fisher Scientific). For data processing, both Xcalibur and Compound Discoverer 3.3 software (Thermo Fisher Scientific) were used.

#### 4.1.6. Light Microscopy

Retina tissue sections (5 µm) were fixed in 10% neutral-buffered formalin for 6–24 h, stained with hematoxylin and eosin. After fixation, samples were routinely processed using a Tissue-Tek^®^ VIP^®^ (Sakura, Torrance, CA, USA), embedded in paraffin, and 5-μm tissue sections were prepared. Tissue sections were covered with 1 mL FAOD (100 µg/mL; 3.83 U/mL) and were incubated for 3 h at 37 °C. After incubation, the tissue slides were rinsed gently with water. The prepared slides were then stained with hematoxylin and eosin (HE) and were cover-slipped. The total number of drusen that were treated with FAOD was 15. For control treatment, one section was covered with 1 mL 3.83 U/mL FAOD solution and the adjacent section was always used for 1 mL control solution. For this experiment, ultra-thin sections of 2 micrometer were used, to minimize variation.

#### 4.1.7. Statistical Analysis

Statistical data analyses were carried out using GraphPad Prism version 8.4.3. (San Diego, CA, USA). Normality of the obtained data was assessed using the Shapiro–Wilk test. Normally distributed study data were represented as mean ± standard deviation (SD), non-normally distributed data are presented as median with the interquartile range (IQR). For non-normally distributed values, unpaired differences between two groups were assessed using the Mann–Whitney U test and paired differences were assessed using the Wilcoxon matched-pairs signed rank test. Individual comparisons between two groups were performed with paired *t* tests. A *p* value < 0.05 was considered to be statistically significant.

## 5. Patents

“Treatment of diseases with fructosyl-amino acid oxidase” patent filed at the European Patent Office. Priority date: 8 March 2021 EP21161313.

## Figures and Tables

**Figure 1 ijms-25-04779-f001:**
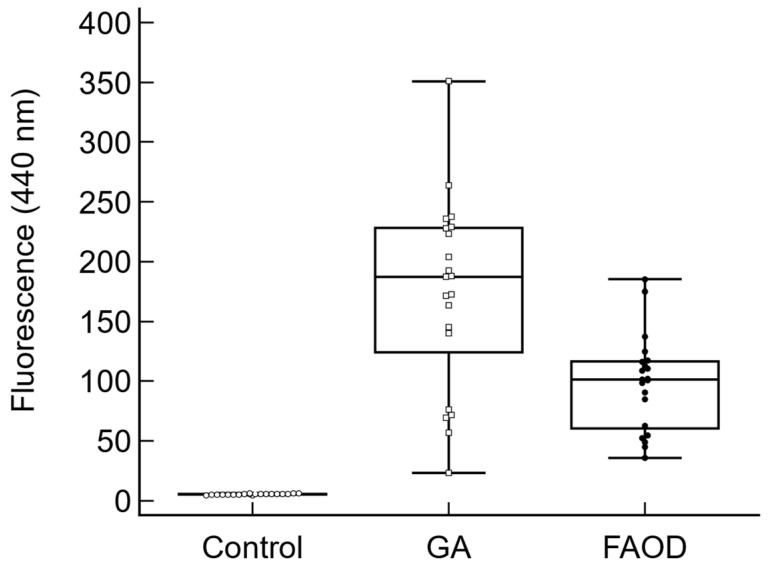
Effect of FAOD treatment on glycated human retina tissues (*n* = 20). The Y-axis shows the autofluorescence (440 nm) values (arbitrary units) of the baseline tissue (base), the glycolaldehyde-treated tissue (GA), and the GA-modified tissue treated with FAOD (FAOD) (*p* = 0.008).

**Figure 2 ijms-25-04779-f002:**
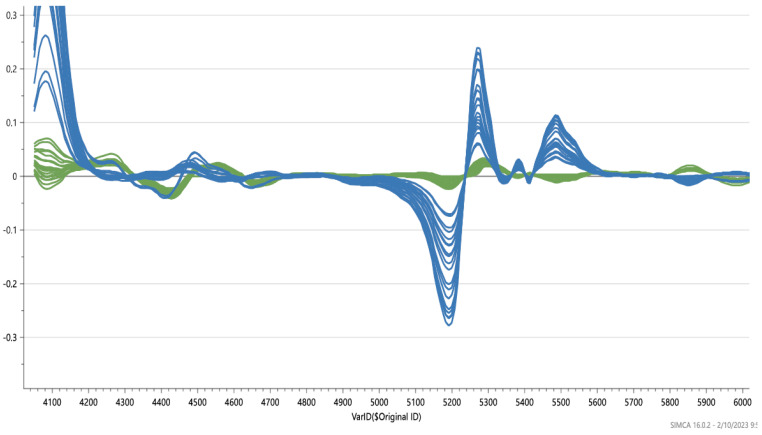
Near-infrared microspectroscopy (wavenumber range: 4100–8000 cm^−1^) on stained tissue sections of human retina. Control (green) vs. FAOD-treated (blue).

**Figure 3 ijms-25-04779-f003:**
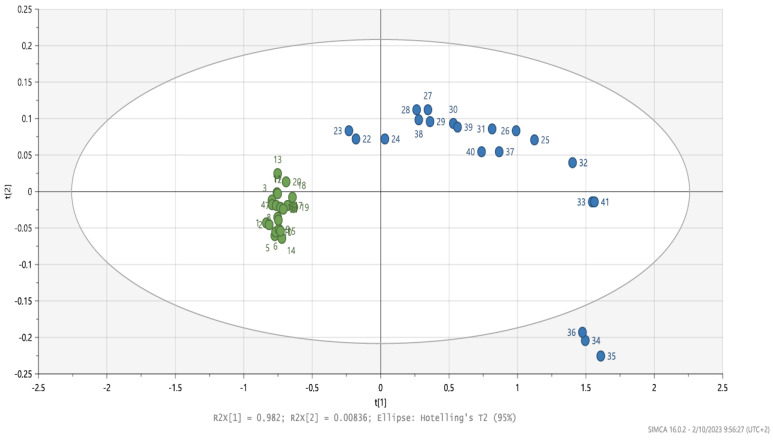
Hotelling plot (wavenumber range: 4100–8000 cm^−1^) on stained tissue sections of human retina. Control (green) vs. FAOD-treated (blue).

**Figure 4 ijms-25-04779-f004:**
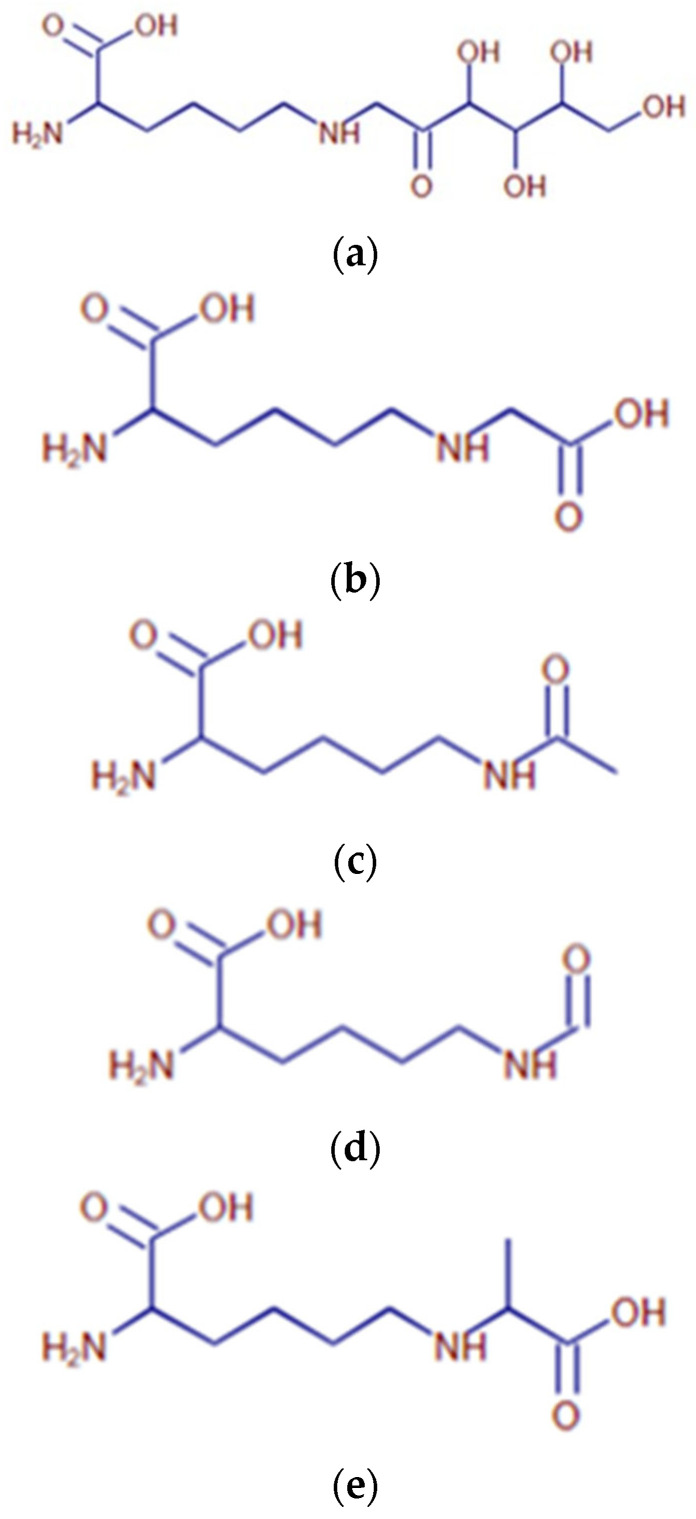
Overview of major lysine-based structures found in the sample. (**a**) Proposed structure for the compound with *m*/*z* 309.16554 at RT 0.85 (Compound # L1), detected in lysine-containing samples. (**b**) Proposed structure for the compound with *m*/*z* 219.13387 at RT 0.89 (Compound # L2), detected in lysine-containing samples. (**c**) Proposed structure for the compound with *m*/*z* 205.11825 at RT 0.91 (Compound # L3), detected in lysine-containing samples. (**d**) Proposed structure for the compound with *m*/*z* 189.12335 at RT 0.96 (# L5), detected in lysine-containing samples. (**e**) Proposed structure for the compound with *m*/*z* 175.10778 at RT 0.98 (# L6), detected in lysine-containing samples. (**f**) Proposed structure for the compound with *m*/*z* 219.13388 at RT 1.07 (Compound # L10), detected in lysine-containing samples. (**g**) Proposed structure for the compound with *m*/*z* 189.12332 at RT 1.09 (Compound # L11), detected in lysine-containing samples.

**Figure 5 ijms-25-04779-f005:**
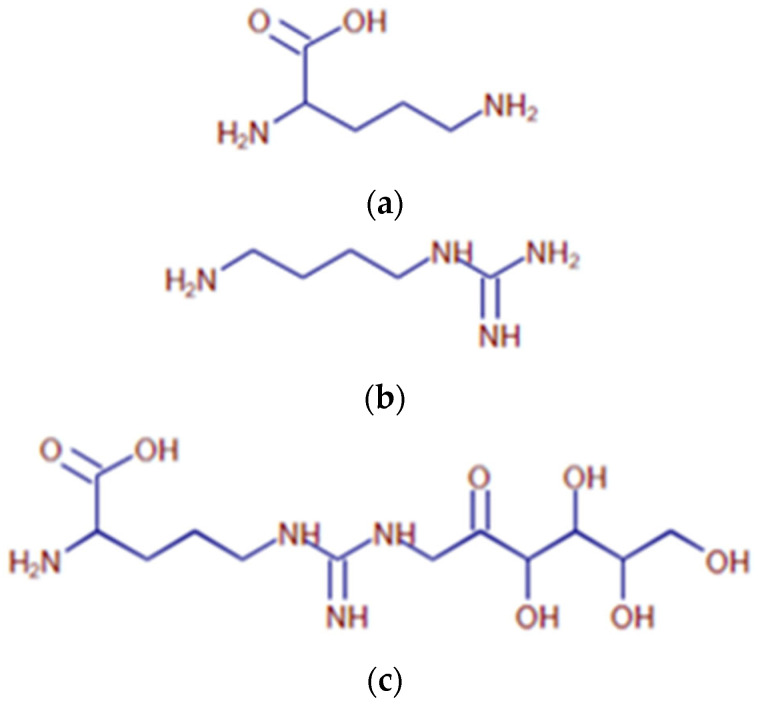
Major arginine-based AGEs found in the sample. (**a**) Proposed structure for the compound with *m*/*z* 133.09695 at RT 0.82 (Compound # A1), detected in arginine-containing samples. (**b**) Proposed structure for the compound with *m*/*z* 131.12904 at RT 0.83 (Compound # A2), detected in arginine-containing samples. (**c**) Proposed structure for the compound with *m*/*z* 337.17098 at RT 0.88 (Compound # A3), detected in arginine-containing samples. (**d**) Proposed structure for the compound with *m*/*z* 319.16064 at RT 0.88 (Compound # A4), detected in arginine-containing samples.

**Figure 6 ijms-25-04779-f006:**
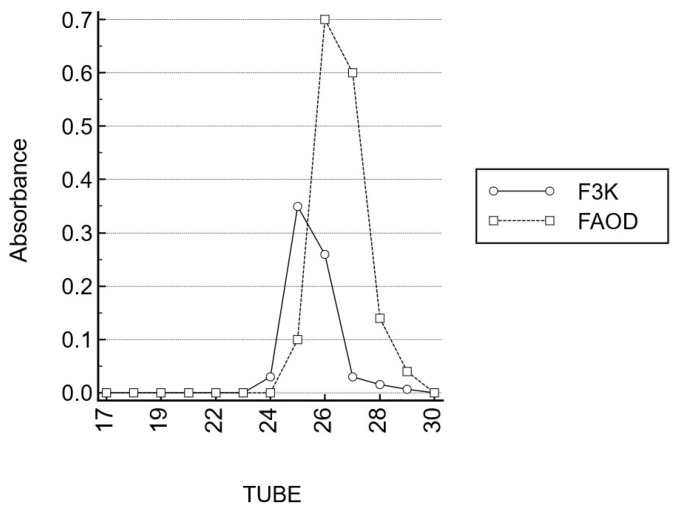
Gel permeation chromatography (Sephadex G-25) of FAOD and FN3K digests.

**Table 1 ijms-25-04779-t001:** Overview of the major characterized compounds in lysine-containing samples.

#	RT (min)	[M + H]^+^ *m*/*z*	Molecular Formula	Annotation
L1	0.85	309.16554	C_12_ H_24_ N_2_ O_7_	Fructosyl-lysine/Glucosyl-lysine
-	0.85	147.11260	C_6_ H_14_ N_2_ O_2_	Lysine
L2	0.89	219.13387	C_9_ H_18_ N_2_ O_4_	Carboxyethyllysine
L3	0.91	205.11825	C_8_ H_16_ N_2_ O_4_	Carboxymethyllysine
L5	0.96	189.12335	C_8_ H_16_ N_2_ O_3_	Acetyllysine
L6	0.98	175.10778	C_7_ H_14_ N_2_ O_3_	Formyllysine
L10	1.07	219.13388	C_9_ H_18_ N_2_ O_4_	Carboxyethyllysine
L11	1.09	189.12332	C_8_ H_16_ N_2_ O_3_	Acetyllysine

**Table 2 ijms-25-04779-t002:** Overview of relevant compounds (possibly AGEs) detected in arginine-containing samples.

Compound #	RT (min)	[M + H]^+^ *m*/*z*	Assigned Molecular Formula
A1	0.82	133.09695	C_5_ H_12_ N_2_ O_2_
A2	0.83	131.12904	C_5_ H_14_ N_4_
-	0.86	175.11854	C_6_ H_14_ N_4_ O_2_
A3	0.88	337.17098	C_12_ H_24_ N_4_ O_7_
A4	0.88	319.16064	C_12_ H_22_ N_4_ O_6_

**Table 3 ijms-25-04779-t003:** Comparative analysis of MS FAOD and F3K digestion. Disappearing AGEs (A = arginine-based, L = lysine–based).

#	[M = H] *m*/*z*	Molecular Formula	Annotation	FAOD	F3K
A1	133.09695	C_5_H_12_N_2_O_2_	ornithine	STRONG	WEAK
A3	337.17098	C_12_H_24_N_4_O_7_	Fructosyl-arginine/glucosylarginine	STRONG	NOT ACTIVE
A4	319.16064	C_12_H_22_N_4_O_6_	Imidazolone A	STRONG	NOT ACTIVE
L1	309.16554	C_12_H_24_N_2_O_7_	Fryctosyl-lysine/glucosyl-lysine	STRONG	WEAK
L2	219.13387	C_9_H_18_N_2_O_4_	carboxyethyllysine	STRONG	WEAK
L3	205.11825	C_8_H_16_N_2_O_4_	carboxymethyllysine	STRONG	WEAK
Formed products					
A9	351.15047	C_12_H_22_N_4_O_8_	Product from arginine, 3-deoxyglycosone and glyoxal	VERY STRONG	Not detected
A2	131.12904	C_5_H_14_N_4_	agmatine	STRONG	Variable (depending on starting mixture)

## Data Availability

The data that support the findings of this study are available from the corresponding author, upon reasonable request.

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
