# Peer review of "Fructosyl Amino Oxidase as a Therapeutic Enzyme in Age-Related Macular Degeneration"

_ijms, 2024, doi:10.3390/ijms25094779_

Round 1
Reviewer 1 Report
Comments and Suggestions for Authors
In general, the text is difficult to read. The manuscript must be significantly improved in writing, making sentences clear and easy to read. The current version is in many places confusing.
The abstract needs to be restructured, to many details of results are reported, methods are missed, and conclusions are not very explanatory.
Also, the introduction is confused, the sentences are not connected to each other and is missed of relevant informations (for example, in lines 39-40 the authors mention a generic treatment for wet AMD without references).
Methods: it is not explained the reason why the experiments are performed in human and porcine retina.
The haematoxylin method reported in fig.4 is not present in method section.
Results: it is impossible to recognize the retinal tissue in Figure 4. In the paraffin sections stained with haematoxylin the retinal layers are not present and the magnification scale is missed.
Minor concerns:
There are some typos need to be corrected.
Some examples:
Line 114: retinav
Lines 120-121: “has reduced by reduced”
Line 127: reported instead of depicted
Line 133: Agmatine (4-aminobutyl-guanidine) was formed, what does it means?
Author Response
REVIEWER 1
In general, the text is difficult to read. The manuscript must be significantly improved in writing, making sentences clear and easy to read. The current version is in many places confusing.
The abstract needs to be restructured, to many details of results are reported, methods are missed, and conclusions are not very explanatory.
The text has been revised. The abstract has now been restructured. Sentences of the introduction have been shortened in the revised text. The method description has been enlarged, and the conclusions have been expanded in the revised text. Extra clarifications have been provided. It should be noted that the format of the abstract does not allow to provide a lot of details.
Also, the introduction is confused, the sentences are not connected to each other and is missed of relevant informations (for example, in lines 39-40 the authors mention a generic treatment for wet AMD without references).
Connections between sentences have been improved. A new reference regarding the generic treatment of wet AMD has been added
Methods: it is not explained the reason why the experiments are performed in human and porcine retina.
The reason why both human and porcine samples have been studied in this manuscript is because of the limited availability of human tissues. Human eye tissues, originating from older donors, have the advantage that naturally formed drusen have been studied. As the availability of human cadaver eyes is limited (the ethical committee only gave the permission to use cadaver eyes which have been rejected for corneal transplantation), the number of retina experiments needs to be increased with animal retinas to provide more evidence. In contrast, porcine retinas are relatively easy to obtain (waste product of slaughter houses), and allow to study chemically indued AGES (glycolaldehyde treatment). This has now been clarified in the revised version of the manuscript (at the end of the introduction section).
Reviewer 2 Report
Comments and Suggestions for Authors
Age-related macular degeneration is a leading cause of vision-loss worldwide. The pathophysiology is complex but involves the formation of drusen that ultimately can progress to advanced atrophic or neovascular forms. AGEs are implicated in the pathophysiology of AMD. The authors of this study attempt to identify a novel therapeutic solution by studying FAOD in the context of AMD. Overall, the study is well-controlled, and interpretations are appropriately stated. This reviewer commends the authors for acknowledging the limitations of this study. This reviewer has one major and a few minor critiques, as noted below.
Major:
-Perhaps the authors might consider an in vivo study to test the safety of FAOD in mice. A potential experimental design might be intravitreal injections of FAOD vs. vehicle control and measuring ERG responses to determine possible effects on the retina. This would help enhance the translation of this study to human disease.
Minor:
-Could the authors make dot plots representing individual data points used to create Figure 1?
-Could the authors elaborate on the feasibility of using FAOD as a potential therapeutic choice? Is this reagent being used systemically in other human diseases? My concern is regarding the negative off-target effects this reagent might have on the neuronal retina (see Major #1).
Comments on the Quality of English LanguageSome minor grammatical errors were detected by this reviewer. The errors did not take away from the content of the study. Editing during revisions or the proof stage would be sufficient to correct.
Author Response
REVIEWER 2:
Age-related macular degeneration is a leading cause of vision-loss worldwide. The pathophysiology is complex but involves the formation of drusen that ultimately can progress to advanced atrophic or neovascular forms. AGEs are implicated in the pathophysiology of AMD. The authors of this study attempt to identify a novel therapeutic solution by studying FAOD in the context of AMD. Overall, the study is well-controlled, and interpretations are appropriately stated. This reviewer commends the authors for acknowledging the limitations of this study. This reviewer has one major and a few minor critiques, as noted below.
The limitations are mentioned in the manuscript/
Our study is hampered by a number of limitations. First of all, experiments in this study have only been performed on in vitro or ex vivo material. Human clinical trials are indispensable for assessing the clinical validity of our findings. It can be anticipated that in the in vivo situation, in order to obtain optimal results, multiple treatment rounds will be needed. Besides, it might as well be that after time in humans, protein crosslinking recurs and treatment should be repeated. Finally, the power of our study is hampered by a low number of human eyes.
Major:
-Perhaps the authors might consider an in vivo study to test the safety of FAOD in mice. A potential experimental design might be intravitreal injections of FAOD vs. vehicle control and measuring ERG responses to determine possible effects on the retina. This would help enhance the translation of this study to human disease.
We have planned additional studies on toxicity. Viability of retinal cells have been tested using an Ussing chamber. We could not observed any loss of viable cells upon FAOD exposure (these expected therapeutic concentrations are in the nmol/L range and thus very low). We thank the reviewer for the useful suggestion.
Minor:
-Could the authors make dot plots representing individual data points used to create Figure 1?
We have now adapted figure 1. All individual dots can now be seen on the three box-and-whisker plots.
-Could the authors elaborate on the feasibility of using FAOD as a potential therapeutic choice? Is this reagent being used systemically in other human diseases? My concern is regarding the negative off-target effects this reagent might have on the neuronal retina (see Major #1).
So far, FAOD has only been therapeutically used in dermatological diseases (reference De Decker I, Notebaert M, Speeckaert MM, Claes KEY, Blondeel P, Van Aken E, Van Dorpe J, De Somer F, Heintz M, Monstrey S, Delanghe JR, Enzymatic deglycation of the skin by means of combined use of fructosamine-3-kinase and fructosyl amino-acid oxidase. Int J Mol Sci 2023 24: 8981. doi: 10.3390/ijms24108981. ). This reference is included in the manuscript as reference number 21.
As this treatment is brand new, it has only been used in preliminaru pre-clinical and clinical studies (not systematically). At the time being, there is no GMP produced enzyme available, which limits the clinical use in human patients to dermatogical topical applications.
The risk of toxicity is extremely low. Due to the enzymatic character of the FAOD treatment,
one enzyme molecule may hits millions of target molecules (the turnover number of the enzyme, which has a relatively small molecular mass, is high).
We already have carried out a first series of toxicity studies on viability of retinal cells which showed no toxicity at all.
Comments on the Quality of English Language
Some minor grammatical errors were detected by this reviewer. The errors did not take away from the content of the study. Editing during revisions or the proof stage would be sufficient to correct.
The text as been checked. A number of typos have been corrected.
Round 2
Reviewer 1 Report
Comments and Suggestions for Authors
The authors ignored my suggestion to revise the introduction, and the abstract has barely changed from the previous version.
I find it hard to believe the authors' claim that the drusen are reduced, based on the results they show. I do not understand how they can make such an assertion.
In my view, this paper is not ready for publication in its current form.
Author Response
- please see the attachment

Round 3
Reviewer 1 Report
Comments and Suggestions for Authors
I will try to better express my disbelief, which does not refer to all the results presented, but to the two example images of retinas. The drusen in a retina can have different sizes, so to do an accurate study, relating to size, a very large number of samples would be needed and the authors themselves state that to complete the studies it would be necessary to increase the number of samples. Therefore, in my opinion, given that it has been demonstrated with multiple techniques that treatment with FAOD reduces drusen, size analysis can be excluded or could be included as supplementary material.
Author Response
I will try to better express my disbelief, which does not refer to all the results presented, but to the two example images of retinas. The drusen in a retina can have different sizes, so to do an accurate study, relating to size, a very large number of samples would be needed and the authors themselves state that to complete the studies it would be necessary to increase the number of samples. Therefore, in my opinion, given that it has been demonstrated with multiple techniques that treatment with FAOD reduces drusen, size analysis can be excluded or could be included as supplementary material.
We thank the reviewer for this constructive comment. To improve clarity, we have modified the description as follows:
Retina tissue sections (5 µm) were fixed in 10% neutral-buffered formalin for 6-24 h
stained with hematoxylin and eosin. After fixation, samples were routinely processed using a Tissue-Tek® VIP® (Sakura, Torrance, CA, USA), embedded in paraffin, and 5-μm tissue sections were prepared. Tissue sections were covered with 1 mL FAOD (100 µg/mL; 3,83 U/mL) and were incubated for 3h at 37°C. After incubation, the tissue slides were rinsed gently with water. The prepared slides were then stained with hematoxylin and eosin (HE) and cover-slipped. The total number of drüsen that were treated with FAOD was. 15. For the control treatment, one section was covered with 1 mL 3.83 U/mL FAOD solution and the adjacent section was always used for 1 mL control solution. For this experiment, ultrathin sections of 2 micrometer were used, to minimize variation."
Furthermore, we have now added the number of investigated drusen to the text and have moved the former Figure 4 to the supplementary material.
The changes have been marked in yellow.